# The spatial configuration of biotic interactions shapes coexistence-area relationships in an annual plant community

David García-Callejas [1,2 ✉], Ignasi Bartomeus [1] & Oscar Godoy [2]

The increase of species richness with area is a universal phenomenon on Earth. However, this observation contrasts with our poor understanding of how these species-area relationships (SARs) emerge from the collective effects of area, spatial heterogeneity, and local interactions. By combining a structuralist approach with five years of empirical observations in a highly-diverse Mediterranean grassland, we show that spatial heterogeneity plays a little role in the accumulation of species richness with area in our system. Instead, as we increase the sampled area more species combinations are realized, and they coexist mainly due to direct pairwise interactions rather than by changes in single-species dominance or by indirect interactions. We also identify a small set of transient species with small population sizes that are consistently found across spatial scales. These findings empirically support the importance of the architecture of species interactions together with stochastic events for driving coexistence- and species-area relationships.

[1] Estación Biológica de Doñana, C/Américo Vespucio 26, 41092 Seville, Spain. [2] Departamento de Biología, Instituto Universitario de Investigación Marina (INMAR), Universidad de Cádiz, E-11510 Puerto Real, Spain. ✉email: josedavid.garcia@uca.es

One of the few general patterns observed in nature is the positive relationship between species richness and spatial extent, formalized as the species–area relationship (SAR)[1,2]. The universality of these positive relationships and their canonical form across taxa and habitats suggest that there are common processes shaping the spatial distribution of biodiversity[3]. After almost a century of studies on SARs since the first formalizations by Arrhenius[1] and Gleason[4], different branches in ecology and biogeography have explored several specific mechanisms potentially shaping SARs (e.g., niche differentiation, colonization-competition trade-offs, dispersal limitation). These deterministic processes overall maintain species with different environmental niches, thanks to the increase in environmental heterogeneity with spatial extent[5–7]. Likewise, SARs are modulated by the simple fact that larger areas harbor higher species richness and population densities, or larger areas have by definition a higher probability of receiving immigrants by dispersal or stochastic events[8]. The relative importance of deterministic processes and stochastic/sampling events in driving SARs has been explored in different contexts with contrasting results[3,7,9,10]. Despite these recent efforts, we still poorly understand how SARs emerge in a more general way from integrating the collective effects from multiple deterministic processes and stochastic factors.

A straightforward way to make solid progress into this aim would be to explicitly account for each of all potential deterministic and stochastic sources of SARs. However, such an approach is so far lacking due to theoretical and practical reasons. As an alternative, it has been recently suggested that this integration can be summarized by understanding how variation in local patterns of species coexistence across scales determine the shape of SARs[11]. The underlying argument is that multiple deterministic processes and stochastic events jointly modulate the spatial location and relative abundance of species across space, through their influence on species interactions and coexistence patterns. Although there are multiple available methods to study coexistence patterns, we focus on a structuralist approach because it provides theory and associated toolboxes that unify the deterministic nature of ecological processes and the intrinsic variability of real-word systems[12,13]. Specifically, we define coexistence as the ability of two or more species to persist (i.e., maintain positive abundances in the long term) within ecological communities despite showing contrasting population trajectories. Following a structuralist approach, this coexistence condition is met when the architecture of species interactions creates a domain of opportunities for species to coexist (i.e., feasibility domain) that is able to accommodate differences in their intrinsic growth rates[14,15]. A structural analysis, by itself, does not conclude whether such coexistence is dynamically stable or not in the face of perturbations, but it has been argued that these conditions are tightly related[16]; therefore in this study we take the feasibility of a community as a proxy for its coexistence, and from here onwards we refer to species coexistence in the sense of feasible combinations of species[14]. In multispecies communities, intra and interspecific (both direct and indirect) interactions can increase or decrease the size of the feasibility domain with the general prediction that the larger this domain is, the larger the probability of persistence of individual species and of the entire community[14]. The empirical study of SARs through the lens of a structuralist approach provides a platform for quantifying two assessments that remain unexplored. First, it allows quantifying the relative importance of passive sampling and stochastic events coming from increases in spatial extent (Fig. 1a) versus environmental heterogeneity (Fig. 1b) in shaping the increase of the number of coexisting species with area. Second, the possibility to distinguish which species coexist at increasing spatial scales from those which

do not (i.e., transient species defined as sink species with a viable population source elsewhere, see Fig. 1c) allows exploring the role of sink-source dynamics for shaping SARs.

Here, we empirically assess the effects of species persistence on driving SARs. We specifically ask three questions: (1) How important are increases in environmental heterogeneity versus increases in spatial extent for determining species persistence? (2) What is the relative contribution of direct and indirect interactions to species persistence across spatial scales? and finally, (3) How do SARs emerge from the balance between persisting and transient species? To address these questions, we performed detailed field observations of fecundity and neighborhood composition of 9130 individuals from 19 annual plant species during 5 years (2015–2019) and across different spatial scales (9 plots of 72 m², giving a total sampling area of 650 m²) in a grassland community in which we know that the performance and spatial location of these annual plant species is affected by a marked gradient in soil salinity[17]. With this empirical information, we estimated the strength of species interactions (both competition and facilitation) and their intrinsic ability to grow following mathematical models that describe reasonably well the population dynamics of interacting species. Finally, we assessed which species are predicted to persist, that is, which species or combinations of species (if any) possess a configuration of interactions that accommodate differences in their intrinsic growth rates. To answer the first question, model parameterization and predictions of species persistence were done for two contrasted environmental scenarios, namely homogeneous or heterogeneous environments, by allowing models to account for spatial variability at the plot scale. To answer the second question, we computed the simplest community (defined by the number of species) allowing the persistence of each individual species. To answer the third question, SARs were built by accounting for the number of species predicted to persist across spatial scales (see Fig. 1 for a schematic diagram).

## Results

In line with theoretical expectations, the relationship between the predicted number of coexisting species with area, the coexistence–area relationship (CAR)[11], matches the shape of classic SARs (Fig. 2), with power-law relationships whose exponents ($z = 0.21$ to $0.45$, Supplementary Table 2) are well within the range of slopes observed for standard SARs[18]. But contrary to these expectations, we find that despite the marked soil salinity gradient present in the study area, increases in spatial extent without accounting for environmental heterogeneity explain most of the coexistence predicted in our system (Fig. 2). In particular, the contribution of spatial heterogeneity to species coexistence is ~12% when averaged across all years and spatial scales. The relationships obtained in Fig. 2 are robust to variations in model coefficients and vital rates (Supplementary Notes 1 and 2).

On average, between 70% and 100% of the species predicted to persist in our system reach coexistence via direct pairwise interactions, whereas coexistence by indirect interactions in multispecies combinations and by switches in local dominance was much less prevalent (Fig. 3). Moreover, the relative importance of coexistence mediated by pairwise direct interactions increased with area and was relatively robust across years, with the exception of 2018, when an extreme flooding event from March–May effectively knocked out almost all plant individuals in the study area. Despite most of the species being predicted to persist in the system, we still identified across spatial scales a small set of transient species that represents on average 13.5% of the plant community (Fig. 3). Although the identity of transient species is variable across years, they all share characteristics

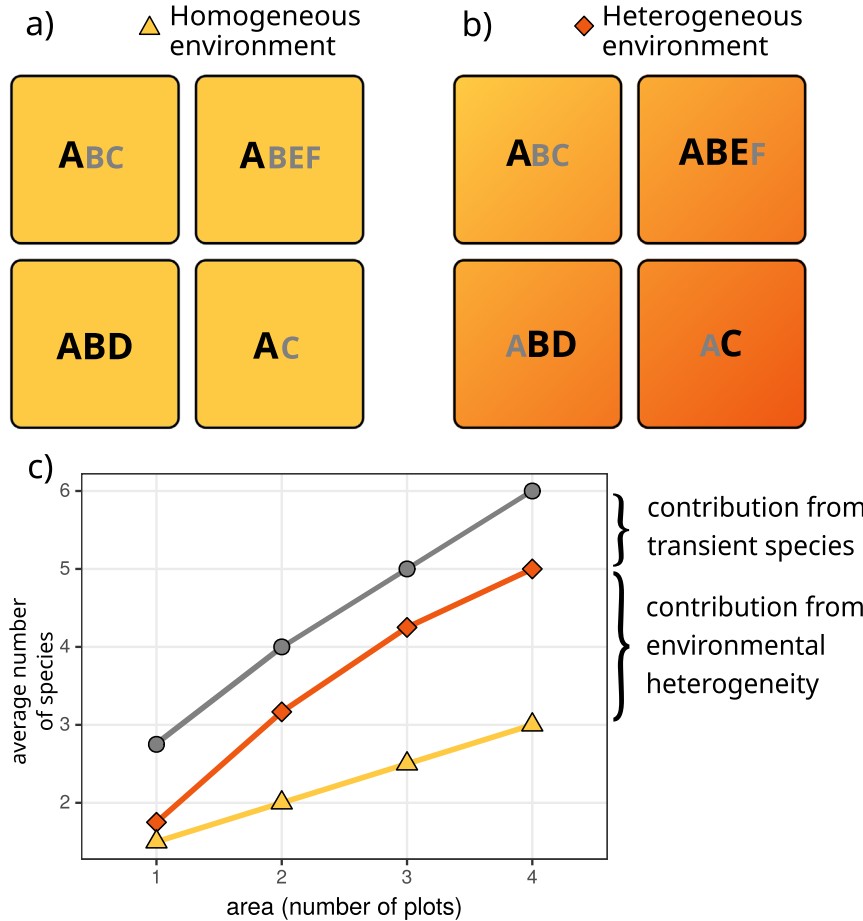

**Fig. 1 A schematic dissection of species–area relationships (SARs) and their relation to species persistence in homogeneous and heterogeneous environments.** In panels (**a**, **b**), each square represents a sampling area, and letters represent species observed in that area. In (**a**) we assume species experience the environment in a homogeneous way, meaning that their intrinsic growth rates and direct and indirect interactions do not change across squares. In (**b**) we consider that species are sensitive to changes in environmental heterogeneity to the extent that it promotes variation in both intrinsic growth rates and their network of interactions across the sampling area. Coexisting species are represented by bold letters, and transient species by smaller gray letters. In (**c**), we show the corresponding coexistence–area relationships for both environments, and the SAR. Variation among these curves can be ascribed to environmental heterogeneity (orange-yellow curves) and the prevalence of transient species (gray-orange curves) in the system. Finally, note that SARs are built by taking the average of all combinations of plots for a given area (i.e., a "Type B" relationship[43]), thus resulting in non-integer values at intermediate areas.

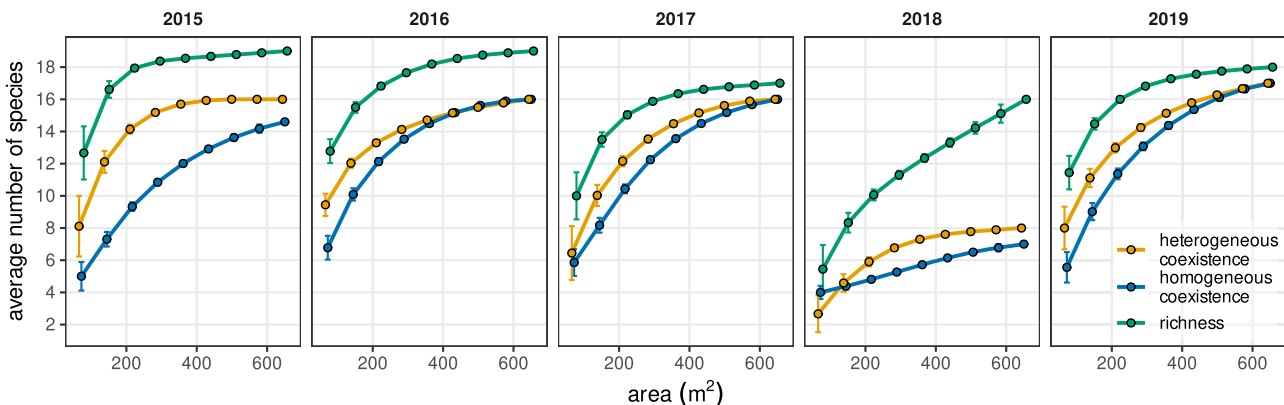

**Fig. 2 Species– and coexistence–area relationships in a Mediterranean grassland community across 5 years.** Points represent average values, and error bars represent the standard error of these averages. Note that curves describing the contribution of homogeneous and heterogeneous environments to the number of coexisting species (blue and yellow lines, respectively), and the curve describing the overall increase of species richness with area (green lines) remain similar across years, except for 2018. That particular year a series of strong rainfall events flooded our system during 2 months at the peak of the growing season (March–May). This extreme event represents a unique opportunity to document how coexistence– and species–area relationships are perturbed and how they can recover afterwards. Points represent average values over all combinations of plots resulting in a given area, as in Fig. 1. $N = 9$, 36, 84, 126, 126, 84, 36, 9, 1 combinations of plots, from the smallest to the biggest area, respectively.

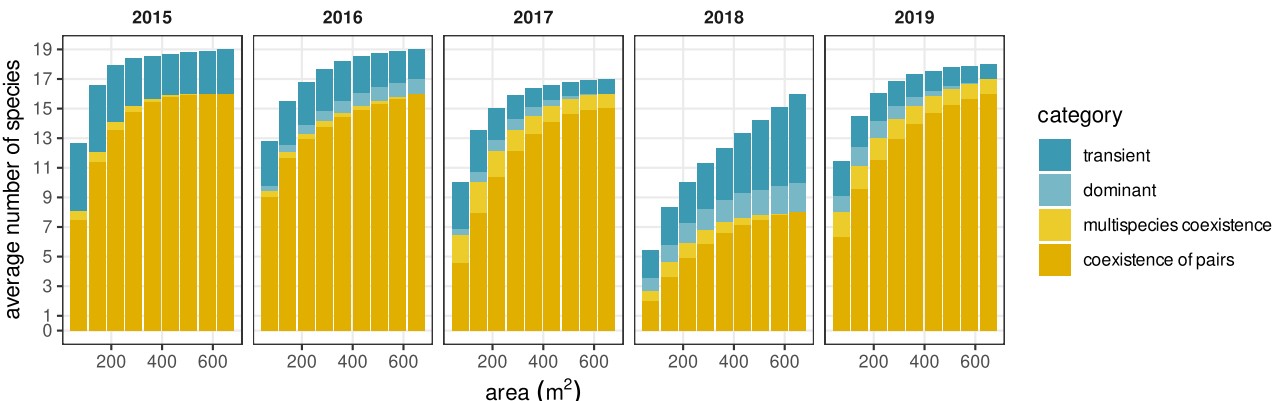

**Fig. 3 Coexistence relationships and their variability with spatial extent.** Relative bar height for each color represents the average number of species in each category when pooled across combinations of plots. These categories detail which specific configuration of species interactions predict their persistence. Configurations of direct interactions that promote persistence of species pairs (category coexistence of species pairs, dark yellow bars) show across years an overwhelming importance for shaping species–area relationships. Note also that species predicted to be transient (i.e., species not predicted to persist, dark blue bars) are a general feature of the system, and their relative proportion increased during the year of the extreme flooding event (2018). A fully detailed explanation for each category describing how species persist under different configuration of species interactions is given in "Methods", under section "Analyzing species persistence".

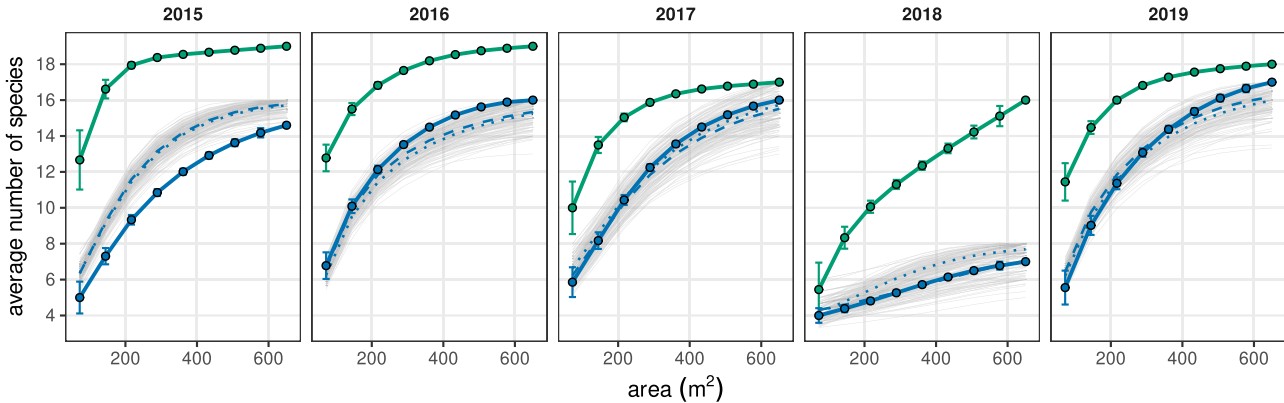

**Fig. 4 Null expectations for coexistence–area curves.** For reference, species–area curves (green lines and associated dots) and coexistence–area curves with homogeneous parameterization (blue lines and dots) are also depicted. Points represent average values, and error bars represent the standard error of these averages. Null replicates are shown in light gray, and averages of these replicates are shown as dotted lines (null model reshuffling all interaction coefficients) and dashed lines (null model keeping intraspecific terms fixed, and reshuffling interspecific terms). $N = 9, 36, 84, 126, 126, 84, 36, 9, 1$ combinations of plots, from the smallest to the biggest area, respectively.

commonly attributed to transient species in the literature[19]. In particular, species systematically classified as transient (e.g., *Suaeda splendens* and *Lythrum tribracteatum*, which show strong preferences for very high soil salinity conditions) were more likely to be comparatively rare (Supplementary Table 3), reinforcing the evidence from our classification based on the ability of species to persist by its local interactions.

Despite the predominant role of direct interactions among species for predicting species coexistence, we interestingly found no significant differences between the CARs derived from two complementary null models and the empirical parameterizations (Fig. 4), In these null models, we reshuffle either the complete set of interaction coefficients in our communities or only the interspecific coefficients, keeping intraspecific ones fixed (see "Methods" for more details). In 2016, 2017, and 2019, the observed coexistence is only marginally higher than that expected from both null models, whereas in 2015 and 2018 the observed coexistence is similar or marginally lower than in the null models. Both null models displayed almost identical average coexistence–area curves in most years, not being able to differentiate between rearranging all coefficients and only the interspecific ones. Overall, these null model

analyses suggest that CARs in our system arise independently from species identity.

## Discussion

Our findings provide empirical evidence that SARs emerge from how local species interactions scale-up to generate coexistence patterns across spatial scales. In particular, we have shown that CARs in a highly diverse Mediterranean grassland entirely composed of annual plant species are consistently similar in shape to classic SARs. But contrary to expectations, we also show that the contribution of spatial heterogeneity to species coexistence is remarkably lower than the contribution from purely spatial processes. Although these results are sensitive to the study system here considered, they provide an empirical evaluation supporting recent hypothesis that there is a scale dependence of diversity maintenance[11].

The overwhelming importance of direct interactions for predicting species persistence, and its increasing role at larger areas, suggests that in our system, the accumulation of species with area does not occur via changes in local dominance (e.g., species

dominant at low salinity versus at high salinity). Rather, we interpret that this accumulation emerges from two complementary effects. The first one is a deterministic effect that allows species to coexist locally, thanks to the structure of pairwise interactions. We know that specific mechanisms such as species differences in resource requirements, in tolerance to soil salinity, and in their mutualistic interactions with floral visitors, promote the coexistence of species pairs in our system via niche differences when intraspecific competition exceed interspecific competition[17]. Nevertheless, we do not discard the additional importance of other mechanisms acting in our system such as more general facilitation effects, which are accounted for in our modeling approach, or natural enemies[20]. The second effect reflects the fact that sampling larger areas increase the chance of more coexisting pairs to co-occur in different places. This latter result suggests that stochastic and dispersal events play an important role in our system[11]. Smaller areas attain in general smaller populations that are more prone to extinction[21,22]. As area increases, the risk associated with negative stochastic effects in smaller areas is reduced because populations are represented by more individuals, which in turn increases the chance of species to interact. Theoretically, an increase in the number of opportunities for species to interact could either promote or hinder their persistence, but we empirically show that the structure of direct interactions between species pairs is key to maintain diversity. Related to this result, a critical question still unanswered is: Over what area do species interactions limit species diversity?[11]. The saturating trend observed across years (except for 2018) of the homogeneous parameterization CARs suggests that the limit to maintain species diversity in our system starts at the largest scale of analyses (650 m²). Below that spatial scale, fewer number of species are predicted to coexist most likely due to a combination of dispersal limitation and stochastic events.

Despite our results suggest a small role of abiotic heterogeneity in driving species persistence, it may still be of importance for determining their relative abundances. For instance, while some generalist species (e.g., *Hordeum marinum*, *Leontodon maroccanus*) attain high abundances across all the study area, it is also possible to observe more subtle changes in community composition and species relative abundances across the soil salinity gradient (e.g., *Scorzonera laciniata* and *Sonchus asper* being more abundant at low soil salinity versus *Chamaemelum fuscatum* and *Salsola soda* at high soil salinity). Finally, the little importance of indirect effects in granting coexistence as we increased the study area is surprising given theoretical expectations from competitive communities[23–25], in which indirect effects are expected to play a large role in shaping coexistence. Rather, our results suggest that the large-scale network structure in our study system hardly includes new opportunities for coexistence that were not contained already in direct pairwise interactions. This aligns well with another line of theoretical and empirical evidence that highlights the critical role of self-limiting processes in stabilizing population dynamics of multiple interacting species[26,27]. Note that we discuss these results based on our analyses of coexistence estimated as feasibility. This approach was taken because prior work has shown that feasibility may be a more relevant property than local stability in seasonal systems[28], where coexistence can be attained without a high local stability under short-term dynamics, as also observed in our study system (Supplementary Note 3).

We also show that despite high predicted levels of species coexistence, a small but sizable set of transient species are systematically identified even at the largest spatial scale of our study (650 m²). Transient species can be interpreted as representing, in open systems like ours, outside contributions to the community

in the form of propagules arriving via dispersal or present already in the system in the form of persistent soil seed banks. Therefore, the observed ratio between the number of coexisting species and the overall species richness can be seen as a proxy to quantify the role of source-sink dynamics in the system[11], as well as the importance of the surrounding habitat in shaping its observed diversity. Specifically, comparing differences observed between CARs and SARs shapes to predictions done with a spatially explicit lottery model (see Fig. 2 from[11]) suggests that our system is governed by short dispersal dynamics, and by species showing on average both low and high spatial autocorrelation in their response to the environmental conditions (years 2017, 2019, versus years 2015, 2016, respectively). We, unfortunately, could not distinguish which group of environmental variables were most affecting species performance. Most likely, these variables differ across species, and future work needs to develop techniques that allow distinguishing what factors drive the sensitivity of species responses to changes in environmental conditions. Furthermore, we assumed that our system presents a soil seed bank across time and space but we did not model explicitly its dynamics across time or space. Nevertheless, our simulations show that the influence of seed survival and germination rates on the observed outcomes is comparatively smaller than variations in intrinsic seed production (Supplementary Notes 1 and 2). Given the species ecology and the environmental characteristics (i.e., annual plant species in Mediterranean-type ecosystems), this assumption is ecologically reasonable, but nevertheless our results must be interpreted bearing these limitations in mind. It has been also hypothesized that CARs should converge with SARs (which include coexisting and non-coexisting species) for sufficiently large areas[11]; this is likely to occur in our system if we increased the spatial scale to include those rare hypersaline patches that would allow the persistence of halophytes (i.e., salt specialists). Regardless of the expected concordance between SARs and CARs at large scales, our results highlight transient species as important features of local ecological communities[19].

All patterns observed, i.e., the shape of the CARs, the greater importance of spatial extent compared to spatial heterogeneity, the prevalence of direct pairwise interactions on promoting species coexistence, and the small but consistent frequency of transient species at increasing spatial scales, are robust across years (Figs. 2 and 3). This is a surprising finding in an annual system given the fact that interaction networks are reassembled and disassembled each hydrological year, under contrasting annual precipitation patterns (ranging from 384 mm in 2019 to 625 mm in 2016). It suggests that key structural features of the system are conserved across years, despite differences in species abundances or the composition of specific plots in the system. Our null model analyses show, specifically, that the identity of pairwise interactions, which drive stabilizing effects among pairs of species, is not a key driver of the spatial coexistence patterns in our system. An explanation for these results is that most interactions, both intra and interspecific, are comparatively weak (Supplementary Fig. 1). In 20–40% of the observed pairwise interactions, intraspecific interactions are nevertheless slightly stronger than interspecific ones, generating coexistence domains that can accommodate small competitive differences between species (Supplementary Fig. 2). In other cases, even if intraspecific interactions do not exceed interspecific ones, coexistence between pairs of species is achieved by weak interspecific facilitation. This quasi-neutral state in the annual assembly of our communities contrasts with prior work positing that the maintenance of high levels of diversity corresponds to a situation in which large stabilizing niche differences can accommodate strong differences in competitive ability between species[29,30], but supports more recent theoretical views of the importance of species' functional

equivalence in driving self-organization patterns in ecological communities[31]. The observed quasi-neutral state surfaces even after the occurrence of extreme climatic events, as shown by the patterns observed in 2019, the year after the flooding event that wiped out most of the studied plant community. This extreme disturbance provided an unexpected and very valuable natural experiment, as it showed how in a single year the community recovered richness and coexistence levels comparable to 2015, 2016, and 2017. Similarly, fast recoveries of SARs have been previously described for lizards[32] and marine fouling communities[33], highlighting the importance of dispersal and population buffers against disturbances (seed banks, in the case of annual plant communities) in recovering canonical SAR shapes. We extend these observations of rapid recovery of community properties to empirical CARs under quasi-neutral dynamics, showing that coexistence mechanisms among pairs and multiple species can rapidly restore high levels of species diversity in natural communities.

For over a century, ecologists have been documenting that the increase of species richness with area follows universal rules. By taking a bottom-up approach where we combine theory and modeling with detailed empirical observations of species interactions at local scales, we highlight that spatial extent can be as important as environmental heterogeneity, and in some cases even more important, to maintain high species richness under quasi-neutral species interactions. More importantly, we provide empirical evidence that SARs in our study system emerge from the collective effect of multiple coexistence mechanisms arising from the architecture of species interactions together with stochastic and dispersal events.

## Methods

**Study system.** We conducted our study in Caracoles Ranch, located in Doñana National Park (SW Spain 37° 04′ N, 6° 18′ W). The study area has a Mediterranean climate with mild winters and an average 50-year annual rainfall of 550–570 mm. Vegetation is dominated by annual grassland species, with no perennial species present. A subtle topographic gradient (slope 0.16%) is enough to generate vernal pools at the lower border of the ranch from winter (November–January) to spring (March–May), while upper parts do not get flooded except in exceptionally wet seasons. In our study, an extreme flooding event occurred during the growing season of 2018. A strong soil salinity–humidity gradient is structured along this topographic gradient.

In September 2014, we established nine plots of 8.5 m × 8.5 m along a 1 km × 200 m area. Three of these nine plots were located in the upper part of the topographic gradient, three at the middle, and the last three at the lower part. The average distance between these three locations was 300 m and the average distance between plots within each location was 30 m (minimum distance 20 m). In addition, each plot was divided into 36 subplots of 1 m × 1 m with aisles of 0.5 m in between to allow access to subplots where measurements were taken (total of 324 subplots). This spatial design was established to parameterize population models including an intrinsic fecundity component and the effect of intra- and interspecific pairwise interactions. Specifically, the core of the observations involved measuring, for each focal individual, per germinant viable seed production as a function of the number and identity of neighbors within a radius of 7.5 cm including individuals of the same species. This radius is a standard distance used in previous studies to measure competitive interactions among annual plant species[29,34], and has been validated to capture the outcome of competition interactions at larger scales (1 m²) under locally homogeneous environmental conditions[35]. From November 2014 to September 2019, we sampled 19 species present in the study area each year. We sampled one individual per subplot for widespread species and several individuals per subplot when species were rare (max. 324 individuals/species). This sampling design ensured that all species were balanced in terms of number of observations, and that we capture the full range of observed spatial interactions among species across the study area. Furthermore, we obtained independent estimates of seed survival and seed germination rates in 2016 (see[17] for details on obtaining these rates). These 19 species belong to disparate taxonomic families and exhibit contrasted functional profiles along the growing season (Supplementary Table 1). The earliest species with small size and open flowers, such as *C. fuscatum* (Asteraceae), peak at beginning of the growing season (February), while late species with succulent leaves, such as *S. soda* (Amaranthaceae) and *S. splendens* (Amaranthaceae), grow during summer and peak at the end of the growing season (September-October). All these species represent up to 99% of plant cover in the study area.

**Estimating species interaction networks and intrinsic growth rates.** We estimated the effect of nearby individuals on individual fecundity via a Ricker model of population dynamics, which allowed us to estimate the strength of positive or negative interactions among pair of species, and therefore, to build a matrix of interactions among species. This approach has been previously applied to study annual plant systems under Mediterranean-type climates[36], and it has also recently been shown to have several advantages compared to other formulations[34]. For example, this model implemented using a negative-binomial distribution for individual fecundities is more flexible in terms of modeling over-dispersion than a Poisson model, while maintaining predictions as positive integers. The model is of the form

$$F_{i,t} = \lambda_i e^{-(\sum \alpha_{i,j} N_{j,t})} \tag{1}$$

where $\lambda_i$ is the number of seeds produced by species $i$ in the absence of interactions, $\alpha_{i,j}$ is the per capita effect of species $j$ over species $i$ (which can be positive or negative, thus allowing both competitive and facilitative effects), and $N_{j,t}$ is the number of individuals of species $j$ within 7.5 cm of the focal individual at timestep $t$. We fitted this model to the empirical data using Bayesian multilevel models with a negative-binomial distribution[34]. For model fitting, we used non-informative priors with MCMC settings of 5000 iterations (of which 2500 were warm-up) and 6 chains. The model was implemented using the brms R package[37]. The effect of changes in environmental conditions on species persistence can be phenomenologically evaluated by allowing models to vary in their estimates of species' intrinsic growth rates and the reorganization of species interactions[38]. In our case, to evaluate the effect of environmental heterogeneity on species persistence (Question 1), we developed two complementary models. In both cases, we modeled the observed viable seed production per individual as a function of the identity and abundance of neighboring species. For the model assuming that plant species interact within a homogeneous environment across plots, we pooled together observations from the whole study area, and allowed the intercept and slope of the relationships to vary across years by including year as a random effect. Thus, the $\lambda_i$ and $\alpha_{i,j}$ values in Eq. 1 vary across years, but are homogeneous for the whole study area. We used the means from the obtained posterior distributions as estimates in the subsequent analyses. For the model that assumes that heterogeneous environments across space and time impact species population dynamics, we included an additional crossed random effect "plot", thus obtaining spatially and temporally differentiated seed production in the absence of neighbors ($\lambda_i$) and interaction coefficients ($\alpha_{i,j}$). Importantly, our modeling approach does not evaluate the magnitude per se of the spatiotemporal variability in our system. It rather evaluates the response of plant species to changes in environmental conditions through their effects on vital rates and interaction coefficients (see[39–41] for similar approaches). Likewise, this approach does not model the spatial dynamics of the community or spatially explicit mechanisms such as dispersal, but rather uses observed spatially explicit associations of individuals to infer their vital rates and interaction coefficients. In the following, we refer to the two developed models as "homogeneous parameterization" and "heterogeneous parameterization", respectively (Fig. 1).

The statistical methodology generates a posterior distribution of estimates for each parameter inferred, i.e., for each intrinsic fecundity rate ($\lambda_i$) and interaction coefficient ($\alpha_{i,j}$). These means, by definition, do not capture the full variability obtained with the statistical model, and may potentially be biased, especially for species pairs that have comparatively few observations. To ensure that our results were not biased by using the posterior mean as a fixed value in subsequent analyses, we replicated our analyses using random samples from the posterior distributions instead of the mean values. We generated 100 random draws from each parameterization and compared the obtained curves to the ones derived from the posterior means (Supplementary Note 1 and Supplementary Fig. 3).

Finally, we assume that the study system presents a rich soil seed bank but we do not explicitly model its direct influence on driving the spatial pattern of species interactions or intrinsic vital rates: rather we use fixed field estimates of seed survival and germination rates in our modeling framework (see section "Analyzing species persistence"). This assumption implies that we cannot evaluate the contribution of a spatially or temporally varying seed bank to the shape of CARs and SARs, which remains an open question for future studies.

**Analyzing species persistence.** To analyze which species are predicted to persist and coexist with others in our system, we built communities based on the species' spatial location. At the smallest spatial scale, given a community of $S$ species observed in the field in a given plot and a given year, we calculated the persistence of each species within every community combination, from 2 species to $S$. Thus, we obtained for each species, plot, and year, two estimates of persistence, one from the homogeneous and another from the heterogeneous parameterization. To scale-up our predictions of species persistence at increasingly large areas, we aggregated species composition and persistence patterns from increasing numbers of plots. We consistently evaluated species persistence using a structuralist approach because prior work has shown it is compatible with the model used to estimate interaction coefficients (Eq. 1)[14]. Specifically, for a given community we first used the strength of sign of intra- and interspecific interactions to compute its feasibility domain (note that the structuralist approach can accommodate different signs in the interaction coefficients). Broadly speaking the feasibility domain is the structural

analog of niche differences, and it represents the possible range of intrinsic species growth rates compatible with the persistence of individual species and of the entire community[14]. Indeed, the larger the feasibility domain, the larger the likelihood of species to persist. Yet, computing the feasibility domain does not tell us which species can persist. To obtain such information, we need to check whether the vector containing the observed differences in intrinsic growth rates between species fits within the limits of the feasibility domain. If so, then all species are predicted to coexist. If not, then one or more particular species is predicted to be excluded (see[14] for a graphical representation).

In order to quantify the feasibility of ecological communities, the intrinsic growth rates and interaction coefficients must be formulated according to a linear Lotka-Volterra model, or an equivalent formulation[14]. We transformed the parameters obtained from Eq. 1 to an equivalent Lotka-Volterra formulation with the following expression (Supplementary Note 2):

$$r_i = log\left(\frac{1 - (1 - g_i)s_i}{g_i}\right) + \lambda_i \qquad (2)$$

where $g_i$ is the seed germination rate of species $i$ and $s_i$ is its seed survival rate. Thus, we quantified the feasibility of our communities using the $r_i$ intrinsic growth rates from Eq. 2 and the $\alpha_{i,j}$ coefficients, which are not modified. For our main analyses, we used empirical estimates of seed survival and germination rates. We further explored the influence of these vital rates in the transformed intrinsic growth rates in Supplementary Note 2 (see also Supplementary Fig. 4 and Supplementary Table 4).

The structuralist methodology further allowed us to dissect which specific configuration of species interactions is behind species persistence in our system (Question 2), among three possibilities: first, a given species may be able to persist by itself, and hinder the long-term persistence of neighboring species (category *dominant*). Second, pairs of species may be able to coexist through direct interactions (category *coexistence of species pairs*). The classic example of two-species coexistence is when the stabilizing effect of niche differences that arise because intraspecific competition exceeds interspecific competition overcome fitness differences[41]. Lastly, species may only be able to coexist in more complex communities (category *multispecies coexistence*)[23], thanks to the effect of indirect interactions on increasing the feasible domain of the community[14]. A classic example of multispecies coexistence is a rock–paper–scissors configuration in which the three species coexist because no species is best at competing for all resources[24,42]. Because species may be predicted to persist under different configurations in a given community, we assigned their persistence category to the simplest community configuration. For instance, if we predicted that a three-species combination coexists as well as each of the three pairs separately, we assigned these species to the *coexistence of pairs* category[26,43]. Finally, if a species is not predicted to persist but it is observed in the system, we classify it as naturally transient, that is, it will tend to become locally extinct no matter what its surrounding community. In order to ascertain our classification of species as transient, we further analyzed whether these species shared ecological traits known to be common to transient species. In particular, a pervasive characteristic of transient species is their comparatively small population sizes. We explored the relationship between our classification as transient and species abundance through a logistic regression with logit link (supplementary Table 3).

In addition to our main analyses, based on the structuralist approach, we explored the local stability[44] of the observed communities, which evaluates their asymptotic response to infinitesimal perturbations, and thus provides a complementary view to the potential coexistence of the system (Supplementary Note 3).

**Species-area and coexistence-area relationships**. To answer Question 3, we obtained standard SARs for each year, by calculating the average diversity observed when moving from 1 plot (72 m²) to 9 plots (650 m²) of our system. In the classification of SAR types proposed by Scheiner et al.[45], the curves from our system are thus of type III-B, i.e., plots in a non-contiguous grid, with diversity values obtained using averages from all possible combinations of plots. Likewise, the yearly CARs were built taking the average number of coexisting species in each combination from 1 to 9 plots. In this case, a species was taken to persist in a given area if it was persisting alone or if it was part of at least one coexisting community within that area. We obtained CARs for the two parameterizations, i.e., assuming homogeneous interaction coefficients and individual fecundity throughout the study area, or explicitly including spatial variability in these terms. We fitted the CARs from Fig. 2 to power-law functions and obtained their associated parameters (Supplementary Table 2) using the mmSAR v1.0 package in R[46]. In the last step of the analyses, to evaluate the role of species identity in driving these empirical fits of CARs, we compared them to two complementary null models that reshuffle the strengths of per capita interactions between species pairs across the interaction matrix. In particular, as baseline we took the CARs from the homogeneous parameterization, in order to have a unique interaction strength value per species pair. In the first null model, and taking the inferred interaction matrix from a given plot and year, we redistributed the pairwise interaction coefficients randomly. That is, we fixed the number of species observed in a certain plot and year, as well as the structure of the interaction matrix, but randomized the magnitude of observed pairwise interactions (both intra and interspecific interactions) in that community.

The second null model is similar, but keeping the diagonal coefficients of the interaction matrix, i.e., the intraspecific terms, fixed. While the first null model accounted for the effect of interspecific competitive responses, as well as self-limiting processes on driving CARs, the second null maintained self-limiting processes fixed by avoiding changes in the diagonal elements of our interaction matrices. We ran 100 replicates of each model for each year, and obtained the average CARs across replicates. All analyses were carried out in R v3.6.3, using packages tidyverse[47] v.1.3.1 for data manipulation and visualization, and foreach[48] v1.5.1 and doParallel[49] v1.0.16 for parallelizing computationally intensive calculations.

**Reporting summary**. Further information on research design is available in the Nature Research Reporting Summary linked to this article.

## Data availability
All data necessary to reproduce the analyses is deposited at Zenodo (https://doi.org/10.5281/zenodo.5390313)[50].

## Code availability
All code necessary to reproduce the analyses is deposited at Zenodo (https://doi.org/10.5281/zenodo.5390313)[50].

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

## Acknowledgements

We thank the Radical Community Ecology group (https://github.com/RadicalCommEcol/) for fruitful discussions. O.G. acknowledges financial support provided by the Spanish Ministry of Economy and Competitiveness (MINECO) and by the European Social Fund through the Ramón y Cajal Program (RYC-2017-23666). This study was funded by Spanish Ministry of Science and Innovation (MICINN) and the European Social fund through the MeDiNaS project (Mechanisms of maintenance of species diversity in variable environments, RTI2018-098888-A-I00).

## Author contributions

O.G. and I.B. built the research infrastructure. D.G.C., I.B., and O.G. conceived the study. D.G.C. designed and performed the analyses with contributions from I.B. and O.G. D.G.C., I.B., and O.G. wrote the paper.

## Competing interests

The authors declare no competing interests.
