## [Peer Review File. · Nature Communications]

Referee #1 (Remarks to the Author):

COMMENT 1: Overall, the authors tell an interesting story about how coexistence properties scale with area, ultimately influencing the SAR. In the end, however, I found myself underwhelmed by the theoretical construct, the empirical demonstration of 'mechanism', and that the dataset, while interesting, was a rather incomplete case study.

First, the theory. The statement by the authors: "Our results provide strong empirical evidence that large areas promote coexistence opportunities that effectively maintain high species richness" seems particularly unsurprising from the context of metacommunity (i.e., spatial) theory. There is a disconnect in community ecology at the moment from those focused on 'modern' coexistence theory, to which this paper adheres, and those focused on spatial ecology (i.e., metacommunities). While the paper here is aimed at the former, and provides some interesting results, the ideas are not at all novel or surprising to the latter group, which takes these issues as 'self-evident'.

For example, in the introductory paragraph, the authors discuss 2 main drivers of SARs. But I believe the discussion of 'passive sampling' confounds many mechanisms that are semi-distinct. To be fair, this has been confused and confusing in the literature for decades, but passive sampling is a purely statistical problem from a species pool. The other mechanisms described, including more resources, etc is something different. Second, I believe the authors have glossed over spatial coexistence theory into two simplistic processes (sampling vs heterogeneity), whereas there are many more processes inherent to spatial coexistence—much of metacommunity ecology deals with these explicitly, including dispersal limitation, Janzen-Connell effects, colonization-competition trade-offs, mass effects and source-sink, frequency dependence and priority effects, spatial grain dependence, etc. etc. These are not all easily lumped into the two spatial processes indicated by the authors. In fact, when reading further, the authors suggest that "for coexistence that would add to passive sampling effects.... Come in the form of multiple coexistence mechanisms such as changes in local dominance... I don't agree these are passive sampling effects. These are spatial coexistence mechanisms, of which there are many. They do differ from heterogeneity, but I don't see them falling into the neat categories as suggested here.

RESPONSE: We thank the reviewer for considering that we provide here a valuable piece of work. Regarding the reviewer's comment that he/she went lost with the theoretical construct defining the conditions for species to coexist, we have rewritten several parts of the text to make them accessible to the general readership. Specifically, we have explained in detail our definition of species coexistence and species persistence (lines 50-52), the underlying motivation by which the study of SAR can be approximated by understanding the effects of species interactions on species coexistence patterns (lines 52-58), the different categories we considered here by which species can coexist (lines 383-405), and how we interpret non-persistent species within a context of sink-source dynamics (lines 229-236).

Furthermore, we agree with the reviewer that the distinction between mechanisms in the SAR literature (and in the ecological sciences in general) is somewhat blurred, with different subfields speaking fairly different languages. In the original submission, we clearly separated many semi-distinct processes driving SAR because we thought a broad audience would follow better the logic of the paper. In the revised version, we have eliminated this clear distinction, and we now say that many deterministic processes drive SARs and these deterministic processes can be further modulated by sampling/stochastic events. Therefore, we present both sources of SARs in a nested approach to evidence that many sources shaping SARs are semi-distinct (both deterministic and stochastic) (lines 35-38).

Finally, we now make explicitly clear that studying each of all drivers of SARs is impossible for logistic and theoretical reasons (line 41-42), therefore, an alternative approximation that has been recently suggested to solve this problem is to investigate their aggregate effect through the way species interact and modulate their ability to persist (Hart et al. 2017). We specifically follow the philosophy of this latter alternative proposing a combination of theory from a structural stability approach with a unique dataset specifically taken to test these ideas. This combination of theory, methods and data has not been explored before, to our knowledge.

References

Hart, S.P., Usinowicz, J. & Levine, J.M. (2017). The spatial scales of species coexistence. *Nature Ecology & Evolution*, 1, 1066–1073.

COMMENT 2: Throughout, but introduced in the 2nd main paragraph, it seems as though the authors are confounding two types of SARs in their exploration of potential mechanisms. This is very common and confusing in the literature, but also quite critical when understanding mechanisms. One type of SAR is an 'island' SAR, where small patches/islands can lead to things like stochastic extinctions and rescue effects. Another type of SAR is a 'nested' SAR where the whole community is embedded in a single landscape, and SARs are measured by increasing the observation window. Here, coexistence mechanisms are the same, but the 'window' of observation changes, leading to different strengths of different mechanisms at different scales. This is what the authors are mostly focusing on in their empirical work, but the explanation of mechanisms seems a bit confounded. In a nested SAR, the factors that influence its shape are the numbers and relative abundances of species, as well as their spatial location; these are what determines the shape of the SAR in the metacommunity.

RESPONSE: We clarify now in lines 66 and following the specific hierarchical approach we follow to understand how SARs emerge from the collective effects of species interactions. We also clarify in the same paragraph, following the reviewer's comment, that the strength of species interactions estimated from the field parameterizations depends on the species spatial location. Further, we make explicit that we consider nested SARs throughout our study (line 410-412). We hope these changes clarifies our approximation to the study of SARs

COMMENT 3: The authors suggest this is a 'mechanistic' understanding of SARs, and certainly one person's mechanism is another person's phenomenon. However, I must say I'm struggling to see exactly how this approach is particularly mechanistic. It's true that coexistence via intra and interspecific effects can be inferred, but this is pretty far from real mechanisms, only those in the 'Chessonian' framework, which is pretty phenomenological. That is, a Ricker model of growth of individuals within a 7.5 cm radius and their random neighbors, but with no information on soil conditions doesn't seem particularly mechanistic.

RESPONSE: We agree with the reviewer that the word mechanistic has a different meaning depending on the reader. For us, the approach presented is mechanistic because we explore the role of species interactions in shaping SARs, but it is true that for others this approach is not mechanistic if it does not incorporate specific drivers such as soil conditions, mutualistic interactions and a long etc. Regardless of the interpretation, we have removed these mentions to a "mechanistic approach" throughout the text. We also explicitly mention in the introduction (lines 58 and following) the two main advantages of using this approach despite not knowing which are the multiple sources that drive the observed structure of species interactions. The first one is to evaluate the role of environmental heterogeneity in shaping SARs. The second one is to evaluate the role of sink-source dynamics. This second advantage as the reviewer says in the next comment (COMMENT 4) is important because: *"this result is pretty interesting as quantifying source-sink relationships has been exceedingly difficult despite the continued interest in them theoretically"*.

COMMENT 4: I understand the transient species, and I think this result is pretty interesting as quantifying source-sink relationships has been exceedingly difficult despite the continued interest in them theoretically. But the explanation and reasoning behind 'dominant' species being somehow not able to coexist via the criteria used seems rather strange to me. If they're dominant, they coexist in a way that differs from what the inference method used here is. That implies the method is wrong or does not capture what really influences these species, not that these are somehow 'different' species. Here, perhaps the problem is that coexistence is defined by response to interactions. But a species that doesn't interact at all, but just 'is' can certainly coexist if it's λ is high enough to make them 'dominant'. As someone who knows and appreciates this sort of literature, but is also rather skeptical that this approach is really mechanistic, this definition of dominance as a species unable to coexist via the given criteria doesn't really provide me with much confidence about the approach.

RESPONSE: We have rewritten and exemplified the different processes by which species can coexist in lines 383-397 of materials and methods. According to our definition of species persistence and species coexistence within a structuralist framework (363-381), a dominant

species is the one that persists in isolation because “it outcompetes the rest of interacting species”. As many papers have shown before, changes in dominance (e.g. one species outcompetes the other species in one place but it is at the same time outcompeted in another place) is a well-known deterministic processes promoting species coexistence at increasing spatial scales when there is 1) environmental heterogeneity and 2) species differ in their environmental requirements.

We respectfully disagree with the reviewer that for a species to be classified as dominant it only involves “having a high lambda” (i.e. showing a high fecundity in the absence of interactions). The structural approach used here, in the same way as previously stated in coexistence theory, says that a dominant species is a combination of high fecundity and low sensitivity to competition (see Godoy and Levine 2014 for details and associated equations). Therefore, to determine whether a species persists in isolation because excludes the rest of competitors we need to evaluate both the lambdas (intrinsic growth rates) and the alphas (strength and sign of species interactions). Accordingly, we clarify in lines 363-381 that species can coexist when their intrinsic growth rates fall inside the feasibility domain shaped by the interaction structure of the community.

Finally, we would like to explain that our study system comprises annual plant species in which plant cover is almost 100% in all plots evaluated. We did not find this particular case the reviewer is suggesting in which individuals grow alone. This could be perhaps the case of annual desert annuals, but definitely it is not our case in marsh grasslands. Even if we were under desert conditions, we would like to remind that species interactions summarize all types of biotic interactions. These might involve density-dependent competition effects but also those driven by natural enemies, mutualistic interactions with above and below ground taxa and a long etc.

References:

Godoy, O. & Levine, J.M. (2014). Phenology effects on invasion success: insights from coupling field experiments to coexistence theory. *Ecology*, 95, 726–736.

COMMENT 5: I do like, however, the partitioning of species coexistence via homogenous and heterogeneous mechanisms. The fact that heterogeneity didn't provide much to the SAR is interesting, but I also think it's a bit of an overstatement here to imply it doesn't matter for SARs. This is a case study—a nice system to be sure, but heterogeneity is in the ‘eye of the beholder’. The fact that it did not contribute to the SAR much here implies to me that even though the authors ‘thought’ the plots were set up in a heterogenous way, the plants themselves didn't see the heterogeneity much (except for a few ‘transient’ species). In this sense, it's a really cool case study, but I believe less general than it could be. We know that the heterogeneity effect on the SAR emerges at larger spatial scales, where species respond to fundamentally different edaphic conditions, etc. thus, the heterogeneity effect observed here would also be scale dependent (i.e., mixing among smaller scales would give less of a heterogeneity effect than among larger scales). Furthermore, at these larger scales, we would likely be able to better explore whether the ‘transient’ species were just sink species with a source elsewhere, as implied by the authors.

RESPONSE: We thank the reviewer for the positive recognition of this partitioning. In our system, previous studies have shown that plant species are sensitive to spatial

heterogeneity e.g. in soil salinity and in pollinator interactions (Lanuza et al. 2018). Therefore, our expectations were that, even though our plots are located in a single habitat, spatial heterogeneity would play a major role in shaping SARs. Indeed, our results open the question of the scales of spatial heterogeneity that shape them (lines 180-182). We have updated the text to be more explicit in the role of species interactions in that scaling (lines 203-208), of transient species (lines 225-246), and to consider additional insights not predicted by theoretical models. For example, note that heterogeneity in soil conditions may drive species performance, but to a degree that does not imply competitive exclusion (lines 210-223). When heterogeneity is considered among contrasting habitats, the results may be indeed different. Nevertheless, we strongly believe that our study fills a much-needed information about SARs and coexistence relationships within large-scale single habitats. Furthermore, our methodological framework gives a first quantification of these effects in terms of species interactions, providing a way for testing the effects of spatial heterogeneity with (comparatively) minimal information and at any spatial scale.

References

Lanuza, J.B., Bartomeus, I. & Godoy, O. (2018). Opposing effects of floral visitors and soil conditions on the determinants of competitive outcomes maintain species diversity in heterogeneous landscapes. *Ecology Letters*, 21, 865–874.

COMMENT 6: In short, it's a nice story, but incomplete in my mind. To me, it presents an interesting 'proof of concept' for looking at spatial coexistence mechanisms.

RESPONSE: We are happy to read that the reviewer considers that our work is interesting, yet we would like to point out that we are not just studying spatial coexistence mechanisms. These have been previously explored in the literature with a diversity of systems including forests and grasslands. Instead, we are the first to show how studying the effect of local species interactions on coexistence we can decompose the effect of environmental heterogeneity and sink-source dynamics on shaping SARs. This information is not yet available in the literature.

Referee #2 (Remarks to the Author):

COMMENT 1: The authors are to be congratulated on an amazing data set and a bold attempt to show how species area relationships can emerge from coexistence mechanisms. However, in some ways it is a little vague on what those coexistence mechanisms are. The storage effect is mentioned explicitly, but mostly the considerations are broader than specific mechanisms, for example, broadly understanding the role of spatial heterogeneity, which is not exactly a coexistence mechanism but a phenomenon that can be the basis of coexistence mechanisms. Beyond the storage effect, words used for mechanisms are “stabilizing pairwise niche differences, changes in species dominance and multispecies dynamics.” Moreover, one must keep in mind that what is addressed here is the species-area relationship on a very small scale, what might potentially be considered just one habitat. It is well-known that species area relationships are highly scale dependent, and indeed context dependent, but you would not know it from many discussions in the literature

including this one. With these concerns aside, the study would indeed be a very important and impressive one if indeed it were shown that the methods were sufficiently rigorous and adequate to the task. I have concerns in these respects, and I hope the authors will do the work to make it clear to readers that their findings are sound.

RESPONSE: We thank the reviewer for providing these nice comments. Overall, we think these are useful suggestions, which we have followed in full detail. Specifically, we have removed all terms that might be confusing and we have rigorously explained and better justified our modelling approach. These specific changes can be read in our next responses to the reviewer. Finally, we agree with the reviewer that our study can be considered just one habitat because we do not include contrasting life forms such as annual plant species and perennial ones. Yet, we think that this is why our study is interesting. Studies like ours are lacking in the literature. Most of the previous work studying SARs have been done over broad scales in which multiple habitats containing species with disparate ecological requirements have been included. Therefore, it is somewhat obvious that environmental heterogeneity is consistently highlighted as a major driver of SARs. By changing the scale, our study conversely says that this relationship can be overstated, and simply increasing in the spatial extent can be more important than previously thought for creating opportunities for species to coexist. Beyond this debate of the relative importance of spatial heterogeneity, we believe we have proposed a robust approach to study this subject in a more systematic way because it is comparable across systems thanks to the estimation of species interactions.

COMMENT 2: One important issue is that many of the references seemingly justifying the methods do not appear to do so, or not clearly, and the authors could be more explicit in the specific justifications. For instance, from lines 60 to 65, I would expect a justification of the key methods used in this paper to understand scale dependence to be found collectively in refs 22 and 9, but it is not clearly there as far as I can see. The statement 397 to 405 in effect saying that feasible equilibrium is globally stable is certainly not correct, and looks like a misunderstanding of ref 21, which does not say that. The statement lines 125-138 that the storage effect can be evaluated by looking at the correlation between intrinsic fecundity and net interaction is not justified by ref 28. There needs to be more reasoning here to get from the procedure in ref 28 to the statement in the text, and it is doubtfully true in general. Also the claim that is such a thing as coexistence through pairwise niche differences is obscure. The theory on multispecies coexistence justifies this only in the two-species case, where it is trivially true, or in the multispecies case only when the sums of relative limitation ratios across multiple species are less than one, as discussed in the recent book chapter (Chesson 2020). Otherwise, indirect effects must necessarily be taken into account. These particular issues must all be clarified for the work to be convincing.

RESPONSE: We apologize for having the impression that the references were not correctly justified. In the revised version we have provided better justifications of our methodological approach or we have removed some of the analyses presented. Specifically, we agree with the reviewer that our original evaluation of the storage effect did not specifically contain the three processes that involve the storage effect. We only evaluated whether there is across species a positive covariance between intrinsic fecundity and intraspecific competition. Accordingly, we have also removed references regarding this topic.

When we refer to the collective behavior of species interactions on shaping SARs, we have now explained that we refer to the fact that species interactions summarize multiple deterministic processes that modulate the ability of species to coexist (lines 42-47).

We have better explained in methods (lines 363-381) what we considered to be a feasible system, and why it is important for our study. We have, nevertheless, changed the terminology throughout the text, highlighting the ecological implications of feasibility for our analyses, to help the readers not familiar with the structuralist approach. In any case, the characteristics to fulfill the feasibility conditions are the same ones as stated in Saavedra et al. (2017).

We finally prefer to not include the reference provided by the reviewer because our analyses have followed a structural stability approach which involves geometric computations. Meanwhile, the reference provided here follows an algebraic approach. There are important different assumptions between both methods which are explained in detail in the appendix 3 of Saavedra et al. (2017). The structural approach inherently considers indirect effects when obtaining multi-species feasibility, and therefore, the opportunities generated by these indirect effects will be reflected in the feasibility of multi-species communities as compared to the feasibility of each species pair independently. We have clarified that in lines 389-391.

We believe these changes present a fully rigorous explanation of the methods used here. We also made the data and code of the analyses fully available to reviewers, and will be deposited in an open repository upon acceptance.

References

Saavedra, S., Rohr, R.P., Bascompte, J., Godoy, O., Kraft, N.J.B. & Levine, J.M. (2017). A structural approach for understanding multispecies coexistence. *Ecol Monogr*, 87, 470–486.

COMMENT 3: The overall method of analysis seems more sound, but even there, a little better explanation would be helpful. Lines 148-150 seem little justified by refs 11 and 14, although perhaps they do not need justification. "Structural stability" here is not used in the sense that it has been traditionally in ecology, and if this is merely feasibility as the text implies elsewhere, then it is not right. After that point in the text, the reasoning becomes somewhat unfocused and it not so clear how the role of spatial heterogeneity is being assessed beyond what can be discerned from local spatial-scale analyses. There also needs to be a more explicit statement about the use of models in the coexistence-area curves. For instance, when discussing "increasingly large areas in the methods," does this mean that model one was not parameterized on the whole system of plots, but just on the area in question? It is also not really clear that the findings primarily represent increased sampling and not spatial heterogeneity, as the models on the scale in question would be expected to incorporate lower-scale effects, as discussed in hierarchy theory.

RESPONSE: We used structural stability in the exact same sense as Saavedra et al. 2017. Actually, we present our methodology with the exact same words as this reference presents in the appendix S3, so there should be no confusion here for further readers. With respect to how we modelled spatial heterogeneity, we have explicitly detailed how we quantify species interactions with/out considering heterogeneity in lines 341-360. In a nutshell, in both the homogeneous and the heterogeneous parameterization more species are included when

increasing the spatial extent, but in the homogeneous parameterization, the interaction coefficients remain similar across the study area. We also explain the construction of species- and coexistence-area relationships in its own methods section (lines 407-434), where we also detail the null model analysis. In that sense, our distinction of spatial homo/heterogeneity is simply driven by the interaction coefficients and intrinsic growth rates of species across space. By averaging interaction coefficients across the whole study area in the homogeneous parameterization, we effectively dilute lower-scale effects into an overall set of coefficients, and therefore we believe this approach does capture the differences between a homogeneous and a heterogeneous (at the plot level) spatial habitat.

Reference

Chesson , P. 2020. Species coexistence. Pages 5-27 in K. S. McCann and G. Gellner, editors. *Theoretical Ecology: concepts and applications*. Oxford University Press, Oxford, U.K.

Referee #3 (Remarks to the Author):

COMMENT 1: García-Callejas et al. have taken a novel approach to study spatial coexistence by combining a five-year experiment with a theoretical framework based on structural stability. I think the main contributions of this paper are on the coexistence-area relationship, an important concept proposed in Hart, Usinowicz & Levine, *Nature E&E* 2017. This paper shows a strong separation between the species-area relationship. This paper also shows that network structures do not seem to be a strong determinant of the coexistence-area relationship. I see a lot of potential in this paper to open a research avenue in studying spatial species coexistence. That being said, I have some concerns that prevent me from being more optimistic.

RESPONSE: We thank the reviewer for considering that our paper has a lot of potential and can open novel avenues of research. The reviewer has provided a useful set of comments and here below we describe how we have addressed them in full consideration.

COMMENT 2: - The differences between coexistent and transient species are purely theoretical. The authors applied the structural stability measure to detect whether species can coexist using field-parametrized models. However, given the large spatial scales, it is unlikely that the parametrized models capture all biotic and abiotic factors that have non-negligible influences on species coexistence. Thus, I don't think the theoretical distinctions necessarily reflect whether species are coexistent or transient in the fields. More direct evidence would be based on the temporal turnovers of species identities. This point is at the center of the whole paper. I strongly urge the authors to provide more justifications to back up this point.

RESPONSE: The reviewer is right in that our classification of species into “coexisting” or “transient” is driven solely by evaluating the conditions for stability according to a structural stability approach. The alternative the reviewer proposes is interesting but it brings other problems. For instance, studying species turnover with five years of data might be limited to detect transient species, and we do not know for instance the overall effect of the seed bank in recovering all studied species. Nevertheless, we believe this is a very valuable comment

and according to it we have now studied whether transient species in our species possess certain characteristics such as low population size that can validate our theoretical predictions. Specifically, to evaluate whether our classification of species as transient reflects these traits, we performed a binomial GLM testing whether our classification of transient vs coexisting is related to the observed abundances (lines 142-146, 399-405, and Table S3). We found that the probability of being classified as transient as per our approach is positively related to their rarity.

COMMENT 3: - I am not sure whether the partitioning of coexistence mechanisms makes sense. The authors claim that "Local coexistence can be achieved through either direct interactions between two species, or through indirect effects mediated by at least a third species" (L150-152). However, two species that coexist with only direct interactions may no longer coexist when we add a third species. In other words, coexistence is a community-level property which makes it tricky to talk about direct coexistence versus indirect coexistence. In addition, I am not sure if the five categories of coexistence mechanisms are mutually exclusive. To be clear, I am not saying that the authors are wrong, but the authors need to make the arguments stronger and clearer to be convincing.

RESPONSE: This is indeed a very interesting comment because it is normally assumed that more complexity leads to more coexistence, but it can be the other case around as the reviewer points out. To clarify this point, we now explicitly say in lines 389-393 that we evaluate whether interactions create novel coexistence opportunities that are not contained in simpler structures. This means that we take a parsimonious approach and we go from simpler to more complex interaction structures. For instance, if we predict that a species triplet will coexist, but also each of the three species pairs that are contained within the triplet will coexist, then these species are assigned to the category "direct coexistence".

Minor comments:

- In general, I feel that this paper can strongly benefit from a careful rewriting. I list my specific comments on writing below.

We detail here below how we have followed each of these comments.

- L15. 'in nature' can be ambiguous. I suggest the authors change the use of 'in nature' across the text.

Corrected, except for the first sentence of the main text, which serves as a broad introduction.

- L15. I am not sure the structural stability approach counts as a mechanistic explanation.

We have eliminated from the text the word mechanistic.

- L25-27. This sentence is unclear. Please specify what "were quickly reestablished".

We have eliminated that sentence from the abstract, due to word limitations.

- L29. "coexistence opportunities" are unclear.

We have removed that term from the text.

- L32-33. I think it might be good to emphasize that the relationship is not only positive but

follows a canonical relationship.

Corrected (line 26).

- L37. I don't think this is a consensus. I am happy to be proven wrong here if the authors can provide some reference to back up this claim.

The introduction has been modified according to this comment and the comments from previous reviewers, so that the distinction between spatial effects is now better explained as a more nuanced nested combination of mechanistic and stochastic processes (lines 31-38).

- L45-49. This sentence is too long. Please break it into 2 or more sentences.

That specific sentence has been completely rewritten.

- L45-53. I am a bit lost in this paragraph. What is the key question that the authors have in mind? Do the authors want to highlight the controversies on determinism vs stochasticity in constructing SAR? Do the authors want to highlight the scarcity of empirical data? I hope the authors consider rewriting this paragraph to smooth the flow. For example, use this paragraph to highlight the gaps this paper wants to address (e.g. line 63-64).

As specified above, the introduction is mostly rewritten to better reflect previous gaps and introduce our approach more smoothly.

- L57. "the collective behavior of these coexistence mechanisms" is unclear. 'collective behavior' has many meanings. Same problem in L263.

We have replaced "collective behaviour" by "collective effects" throughout the text.

- L59-60. I don't think the characterization of the structural stability approach is accurate. This characterization only applies to purely competitive systems. However, the systems that the authors studied have species interactions other than competitive interactions (L95).

The structural stability framework has been successfully applied to different types of interactions, including positive ones (e.g. Rohr et al. 2014, Saavedra et al. 2016). More in detail, the quantification of the feasibility of a system hinges on interaction/vital rate coefficients consistent with a general Lotka-Volterra model or models that can be assimilated to such a general form. The Ricker model is one of such models, and its exponential form allows for both negative and positive interaction coefficients without further modifications.

References

Rohr, R.P., Saavedra, S. & Bascompte, J. (2014). On the structural stability of mutualistic systems. *Science*, 345, 1253497–9.

Saavedra, S., Rohr, R.P., Olesen, J.M. & Bascompte, J. (2016). Nested species interactions promote feasibility over stability during the assembly of a pollinator community. *Ecology and Evolution*, 6, 997–1007.

- L63. "it makes it possible" -> "it is possible".

This sentence has been completely rewritten.

- L67-72. The sentence is too long.

This sentence has been completely rewritten.

- L98. "the strength and sign" is redundant since the sign would be fixed if the strength is fixed.

Corrected.

- L108. What does 'the importance of species identity' refer to?

We have rewritten that sentence.

- L110-111. What is the "community matrix"? Is it the same as the interaction matrix?

We have dropped references to the "community matrix" to avoid confusion.

- L114-115. The concept of CAD should be introduced earlier in the text and in Figure 1.

We introduce this concept in lines 61-62, although for keeping the specific terms and jargon to a minimum specially in the introduction, we first explicitly mention coexistence-area relationships in lines 115-116.

- L128. It would be great to explain what are storage effects in one sentence.

We have removed the spatial storage effect analyses, according to previous suggestions.

- L129. Is the method in Ref 28 used to detect storage effects? I think it would be great if the authors double-check with a new computational method (Ellner et al., Ecology Letters, 2018, 10.1111/ele.13159).

See comment above.

- L131-133: It is very abstract. It would not make sense to non-experts in Modern Coexistence Theory.

See comment above.

- L133-138. It would be great if the authors explain or at least hypothesize why storage effects are not important in their system given that ample empirical evidence has shown the importance of storage effects across many empirical systems. I don't fully understand what differences the authors are suggesting (e.g. heterogeneity magnitudes).

See comment above.

- L175. I am not sure about the "critical role of pairwise niche differences". First, there are facilitative interactions in the system, in which there is no consensus on how niche differences should be defined or quantified. Second, why are the pairwise fitness differences missing?

The reviewer is right in pointing this out. We now refer to niche differences only in mentioning previous studies that serve to understand our conceptualization. We also explicitly specify that positive pairwise interactions may drive coexistence (line 193-194).

- L223-226. I don't think the statements are entirely correct. In what sense does the second reshuffling "maintain similar niche differences" given that the correlation between a_{ij} and a_{ji} is not conserved during reshuffling?

We have almost completely rewritten the section on null models to better clarify them and the insights they provide, in methods (419-434), results (159-167), and discussion (256-263).

- L238. What does "competitive imbalances" refer to? It would be great if the authors keep

the number of terminologies minimal.

We have changed that term to “competitive differences” (261).

- L244. 'emergent neutrality' has a specific theory in theoretical ecology (e.g. Holt, TREE, 2006, 10.1016/j.tree.2006.08.003). Please use another word (such as quasi-neutral in the previous paragraph).

Corrected.

- L247. Please delete 'very' because it does not add anything.

Corrected.

- Extended Data Table 1. I don't think the data can give rise to the shown significant figures. Please also show the statistical errors.

Corrected (See Supplementary Table 1).

- Code. It is great that the authors shared the codes for review. However, I was not able to run the code because the file "toolbox_coexistence.R" seems to be missing.

The reviewer is right, we missed to include that file. The version attached to this submission should be complete.

Reviewer comments

Reviewer #1 (Remarks to the Author):

I was reviewer 1 in the version submitted to Nature, and I must 'thank' the authors for their very thoughtful and careful attention to my own, and the other reviewer comments. The new version is much more convincing and the response to reviewers respectful and meaningful. In all, I feel this is now a well explained and excellent contribution to the field, and would be happy to see it in Nature Communications. My only comment remaining follows:

-Figure 2 and 4. I wonder why the authors have not included variation/error in their presentation of SARs. It looks a bit too simplistic as it is, and I would be much more convinced to see the actual variation in the estimates.

Reviewer #2 (Remarks to the Author):

This is a fascinating exploration of the coexistence-area relationship based on real data from a Mediterranean annual plant community. The basic method is to fit a model for log seed yield as a linear function of species densities for each plot in their study system, and use this model as a basis of predicting whether the species present in a given year are able to coexist with each other. Then pooling the results of over areas, they were able to construct graphs of numbers of coexisting species as functions of area.

They then try to interpret the results with regard to mechanism. As a theoretical exercise based on a particular kind of model, it stands up reasonably well, but in my view there are numerous mismatches with reality, with effects not clearly known. Nevertheless, I believe that they mean that the allocation of cause by mechanism in the real system cannot be taken too seriously.

First the model:

(A) There is no consideration of a seed bank. The real system presumably has a rich seed bank, with seed survival and germination rates varying substantially between species and times, casting doubt on a model that disregards seed persistence in the seed bank. (B) The model is said to apply reasonably to this system, but the reference for that, Mayfield and Stouffer 2017 is not for this system, and does not in fact support this model, as it claims higher order effects have a major influence. (C) The effects of other individuals on fecundity of a given individual are based on an arbitrary circle around a focal individual. Although this might give reasonable fits to data, real interaction kernels in nature cannot be expected to be like this. (D) The model itself incorporates no spatial effects. No dispersal occurs. There is no accounting for spatial structure. Spatial structure cannot affect the dynamics. It only shows up in the results because the species considered in each plot in each year are simply the ones observed to be there. (E) Coexistence is based on feasibility of the equilibrium. There is no mention of checking that the interaction matrix supports stable coexistence. This perhaps stems from downplaying stability of the equilibrium in the structural approach to coexistence, but it is a bad practice. (F) One could of course go into a lot more issues like departures from the reality such as absence of consideration of natural enemies, no consideration of temporal coexistence mechanisms such as have dominated annual plant coexistence studies for arid environments, etc; but my point is not to bring up everything, but to point to a likely sufficient degree of imprecision in the results that such things as partitioning the outcome between mechanisms cannot be done with much confidence.

Second, the mechanisms

Pairwise coexistence is their invention. It is not without merit, but does not stem from Chesson 2000, which does not discuss pairwise coexistence. Instead, Chesson 2000 discusses coexistence of pairs of species, and coexistence with diffuse competition for the many species case. So they should define what they mean by it more clearly in the main text. They allow spatial heterogeneity to contribute to coexistence only through fecundity differences in space. That is far from the only possibility. They have a category of coexistence called "spatial effects," which evidently is not due physical environmental differences in space. They could be clearer on what goes in these

categories.

Third, the estimation

Although statistically sophisticated, statistical sophistication is not without hazards. For instance, it is observational data, and we can expect that local dispersal will make seed distribution patchy. This means that a randomly chosen individual of a species will not in fact have a statistically balanced neighborhood of other species. That leads to a potential bias in the estimation of competition coefficients as has been discussed theoretically by Pacala and Levin 1997. In addition even if estimation is unbiased, the effects on coexistence need not be. For instance, if intraspecific competition is a bit stronger than interspecific competition, as claimed is common in this manuscript, unbiased estimation error will mean in some fraction of cases, the estimate for interspecific competition will be greater than the estimate of intraspecific competition. Thus, even though in reality all species coexist, it will be concluded that some fraction of them do not. They would then be put in the wrong category.

For these reasons, I see the manuscript as an illustration using a theoretical construct partly based on data from a real system, rather than a strong statement about that system.

References not in the manuscript

Pacala, S. W., and S. A. Levin. 1997. Biologically generated spatial pattern and the coexistence of competing species, Pages 204-232 in D. Tilman, and P. Kareiva, eds. *Spatial Ecology: The role of space in population dynamics and interspecific interactions*. Princeton, New Jersey, Princeton University Press.

Reviewer #3 (Remarks to the Author):

I am Reviewer 3 of this paper when it was under consideration at Nature. I thank the authors for the detailed and thorough response to my comments. All my key concerns are addressed. I especially appreciate the new binomial GLM testing to validate the (theoretical) classification of transient versus coexistent species. Overall, the manuscript is much easier to follow than the previous version, where all key concepts/methods clearly defined and justified. I applaud the authors for this nice paper and I believe it will be a valuable contribution to the study of SAR and CAR.

I only have some very minor points:

- Line 14. "contrary to expectation" is unclear.
- Line 20 & line 287. The authors highlighted the importance of "demographic stochasticity" in the Abstract and the importance of "stochastic and dispersal events" in the Discussion. My understanding is that the authors consider these two things identical (while they are not identical in the general literature). Perhaps the authors should pick one terminology to avoid potential misunderstanding?
- Line 52. What does "showing contrasting population dynamics" mean?
- Line 55. Refs 14 and 15 studies the feasibility domain solely with competitive interactions (as the structural niche & structural fitness concepts best apply to purely competitive systems).
- Line 126. Please remove "very" from "very similar".
- Line 144. Please remove "very" from "very high".
- Color usages in the figures are not consistent.
- y-axis labels are not consistent across figures. I prefer "average number of species" in Figure 1C over "species" in Figs 2-4.

Note: In our responses below, the line numbers refer to the submitted version with track changes. We would like to note that we have also updated the associated github repository in order to include the code necessary to reproduce all the supplementary results. Now, all main and supplementary figures and tables can be fully reproduced.

REVIEWER COMMENTS

Reviewer #1 (Remarks to the Author):

I was reviewer 1 in the version submitted to Nature, and I must 'thank' the authors for their very thoughtful and careful attention to my own, and the other reviewer comments. The new version is much more convincing and the response to reviewers respectful and meaningful. In all, I feel this is now a well explained and excellent contribution to the field, and would be happy to see it in Nature Communications. My only comment remaining follows:

-Figure 2 and 4. I wonder why the authors have not included variation/error in their presentation of SARs. It looks a bit too simplistic as it is, and I would be much more convinced to see the actual variation in the estimates.

We thank the reviewer for his/her appreciation and, overall, for his/her useful suggestions, which have greatly improved the manuscript. In this version we have included error bars (giving the standard error of the mean values) in Fig. 2 and in Fig. 4 as the reviewer suggested.

Reviewer #2 (Remarks to the Author):

This is a fascinating exploration of the coexistence-area relationship based on real data from a Mediterranean annual plant community. The basic method is to fit a model for log seed yield as a linear function of species densities for each plot in their study system, and use this model as a basis of predicting whether the species present in a given year are able to coexist with each other. Then pooling the results of over areas, they were able to construct graphs of numbers of coexisting species as functions of area.

They then try to interpret the results with regard to mechanism. As a theoretical exercise based on a particular kind of model, it stands up reasonably well, but in my view there are numerous mismatches with reality, with effects not clearly known. Nevertheless, I believe that they mean that the allocation of cause by mechanism in the real system cannot be taken too seriously.

We thank the reviewer for these comments. Below we address each one in full detail.

First the model:

(A) There is no consideration of a seed bank. The real system presumably has a rich seed bank, with seed survival and germination rates varying substantially between species and times, casting doubt on a model that disregards seed persistence in the seed bank. (B) The model is said to apply reasonably to this system, but the reference for that, Mayfield and Stouffer 2017 is not for this system, and does not in fact support this model, as it claims higher order effects have a major influence. (C) The effects of other individuals on fecundity of a given individual are based on an arbitrary circle around a focal individual. Although this might give reasonable fits to data, real interaction kernels in nature cannot be expected to be like this. (D) The model itself incorporates no spatial effects. No dispersal occurs. There is no accounting for spatial structure. Spatial structure cannot affect the dynamics. It only shows up in the results because the species considered in each plot in each year are simply the ones observed to be there. (E) Coexistence is based on feasibility of the equilibrium. There is no mention of checking that the interaction matrix supports stable coexistence. This perhaps stems from downplaying stability of the equilibrium in the structural approach to coexistence, but it is a bad practice. (F) One could of course go into a lot more issues like departures from the reality such as absence of consideration of natural enemies, no consideration of temporal coexistence mechanisms such as have dominated annual plant coexistence studies for arid environments, etc; but my point is not to bring up everything, but to point to a likely sufficient degree of imprecision in the results that such things as partitioning the outcome between mechanisms cannot be done with much confidence.

We respond point by point following the letter ordination of the comments:

A) We acknowledge that the seed soil bank has not been modelled explicitly, and therefore, we do not know its relative contribution to modulate SARs. Our study assumes, as the reviewer mentions, that the system has a potentially rich seed bank, which likely contributes to maintain species diversity by storing the unfavorable years to growth such as the strong flooding event of 2018. This is what we discuss in lines 290-293. We now explicitly mention in the methods the limitation that the seed bank was not explicitly modelled to explore its contribution to SARs (lines 253-257 and 394-398).

B) We now clarify in the text that the negative-binomial model using a Ricker function is a model that has been previously used in Mediterranean annual plant systems and that it has been used to model annual plant communities without Higher-Order Interactions (HOIs). For that, we cite in these sentences the study by Bimler et al. (2018) (Lines 348-349). Nevertheless, the study of Mayfield et al. (2017) is the reference that best outlines the general advantages of the modelling procedure regardless of the inclusion of HOIs, so we decided to keep that reference as well. Overall, as pointed out in these publications, and quoting Mayfield et al. (2017), the Ricker model can be applied to study annual plants

“for its general biological plausibility and versatility in multi-species communities regardless of taxonomic group and specific fitness estimator”

Furthermore, Mayfield et al. (2017) point out four specific reasons backing the use of a negative binomial statistical framework for analyzing fitness data coming from annual plant communities regardless of the complexity of the model:

- a. *First, the larger the observed fecundity, the greater the potential error associated with its estimate. This is incorporated into the negative-binomial process, since the variance of a negative-binomial distributed variable increases along with its mean by definition*
- b. *Second, real data are often ‘over-dispersed’ and negative-binomial models afford added flexibility relative to a Poisson process, for example, via the introduction of an additional dispersion parameter that can dramatically improve model fit .*
- c. *Third, fecundities are always integers, and negative-binomial processes make explicit predictions for counts, as opposed to treating count observations as if they behaved like continuous variables.*
- d. *Finally, by explicitly predicting integer values, the best-fit parameters for a negative-binomial model can never make biologically unrealistic predictions such as negative fecundities*

The above considerations are explicit in the main text, in lines 348-350, including explicit examples of the model's advantages.

C) We agree with the reviewer that competition among species can follow kernel distributions. For the case of tree species, this is usually modelled as a function of the distance and the size of the neighboring trees (e.g. Canham et al. (2006)). However, such competition kernels are rarely used in observational studies of annual plant species, and it is not the main goal of this work to explore these kernels. Using a fixed distance of 7.5cm is not something that we have decided alone. Rather, this decision is based on a plethora of previous studies that have standardized measurements of focal plant fitness and associated neighborhood at this specific distance. Therefore, we have rather followed a standard procedure to measure interactions among annual plant species (see, for a few recent examples, the studies by Mayfield et al. (2017), Alexander and Levine (2019), Towers et al. (2020)). This also allows explicit comparisons and future meta-analysis with other studies. We would like to remind the reviewer that, as we note in the main text (L327-330), this distance has been experimentally evaluated to predict well the dynamics of interactions and the outcome of competition for annual plants within areas of 1m² (Godoy and Levine 2014). This is exactly the size of our subplots.

Besides the steps taken to ensure standardization and comparison among studies, the competition kernel approach is not feasible in our study system for logistic reasons. We measured approx. 9000 neighborhoods. If we were to include the minimum of three distances around focal individuals, that would render a total of 27.000 neighborhoods, which are unrealistic numbers to explore.

Additional references not in the main text:

Alexander, J. and Levine, J. (2019). Earlier phenology of a nonnative plant increases impacts on native competitors. *Proceedings of the National Academy of Sciences* 116:6199–6204.

Canham, C. D., Papaik, M. J., Uriarte, M., McWilliams, W. H., Jenkins, J. C., Twery, M. J. (2006) Neighborhood analyses of canopy tree competition along environmental gradients in New England forests. *Ecological Applications* 16:540–554

Towers, I. R., Bowler, C. H., Mayfield, M. M. and Dwyer, J. M. (2020) Requirements for the spatial storage effect are weakly evident for common species in natural annual plant assemblages. *Ecology* 101:e03185.

D) The reviewer is correct in that our methodology does not account explicitly for dispersal processes such as migration or directional dispersal. In our case, we are not modelling the dynamics of spatial associations through time as has been done in recent publications (e.g. in Thompson et al. 2020). Rather, we are analyzing the spatial patterns of species interactions across plots (9 in total) and across time (5 years in total), considering both spatially homogeneous or heterogeneous interaction coefficients. We then assess how these spatial patterns affect the predictions of species coexistence at increasing spatial scales. In other words, we do take the spatial associations as given, as the reviewer mentions. Remarkably, despite the temporal variability observed in the spatial association patterns, we observe similar species-area and coexistence-area relationships across years. It is not our aim to model the spatial structure of the system (e.g. seed dispersal kernels) which would be a really interesting objective but it would require a paper on its own. Rather, to reiterate, we take the observed spatial structure of the system each year and infer the spatial scales of coexistence from it by predicting the number of species to coexist at increasing spatial scales. We clarified this explicitly in the main text (lines 379–381).

Additional references not in the main text:

Thompson, P. L., Guzman, L. M., De Meester, L., Horváth, Z., Ptacnik, R., Vanschoenwinkel, B., Viana, D. S., Chase, J. M. (2020). A process-based metacommunity framework linking local and regional scale community ecology. *Ecology Letters* 23:1314–1329

E) We assume the reviewer refers here to stability in the sense of local asymptotic stability, i.e. the capacity of the system to return to equilibrium in response to infinitesimal perturbations. We did not include this additional stability metric in our analyses for two main reasons:

1) It has been shown to be analytically related to feasibility in the structural sense, such that diverse communities (including positive and negative interactions) with feasible but unstable fixed points are, quoting Gibbs et al. (2017) – new reference 16 in main text – , “very unlikely”.

2) It is out of the scope of our study to provide a detailed stability analysis. In that case, neither feasibility on its own or together with local stability would capture the different dimensions of stability in empirical communities (Domínguez-García et al. 2019). Therefore, we restricted our analysis to a commonly-used coexistence proxy. Furthermore, our study system is an annual plant community, for which it is arguably more relevant to evaluate whether interactions allow the persistence of different sets of species, than the capacity of the system to return to equilibrium after a perturbation, which is an unrealistic situation given the yearly “reset” of the system.

We explicitly include this limitation of our study in lines 58–62 of the revised manuscript.

Additional references not in the main text:

Domínguez-García, V., Dakos, V. & Kéfi, S. (2019). Unveiling dimensions of stability in complex ecological networks. *PNAS*, 116, 25714–25720.

F) The reviewer is suggesting to control every single aspect that influences the dynamics of the community. Although this is always desired, there is no such study available in the literature for obvious reasons. We have explored in previous work the role of floral visitors and soil conditions in our system (Lanuza et al. 2018). These results suggest that both control to some extent the structure of species

interactions, but other mechanisms are likely to have significant roles (See Lines 201-206). As we cannot cover all aspects of species interactions at once, our approach in this study is phenomenological and the interaction coefficients summarize all these types of potential effects such as natural enemies, establishment of positive interactions with mutualists and a long etc. Our paper does not follow the standard approach of modern coexistence theory to decompose temporal and spatial mechanisms of species coexistence. Part of that is because that approach can only be done for pairs of species governed by competitive interactions, and here we focus on a multispecies approach in which competition and facilitation are simultaneously occurring. We clarified that our interactions can be positive or negative in line 205 and lines 411-412.

Second, the mechanisms

Pairwise coexistence is their invention. It is not without merit, but does not stem from Chesson 2000, which does not discuss pairwise coexistence. Instead, Chesson 2000 discusses coexistence of pairs of species, and coexistence with diffuse competition for the many species case. So they should define what they mean by it more clearly in the main text. They allow spatial heterogeneity to contribute to coexistence only through fecundity differences in space. That is far from the only possibility. They have a category of coexistence called "spatial effects," which evidently is not due physical environmental differences in space. They could be clearer on what goes in these categories.

We apologize for the lack of clarity with this term. Pairwise coexistence means "coexistence of pair of species" as the reviewer clarifies. We have changed the terminology in the Fig. 3 and throughout the text (e.g. lines 160, 279, 295, 426, 436).

We do not agree with the reviewer that the only way spatial heterogeneity contributes to coexistence is through changes in fecundity of species across sites. Instead, we allow spatial heterogeneity to contribute to coexistence through changes in intrinsic fecundity rates and the sign and strength of species interactions (both intra and interspecific). These are the two components the framework of structural stability uses to assess the feasibility of the system.

Regarding the term "spatial effects", we do not use that term in the manuscript. We refer to either "homogeneous" or "heterogeneous" parameterizations of our models. The rationale for the "heterogeneous" parameterization is that different sources of spatial variability are potentially reflected in spatial variability in 1) intrinsic fecundity rates, 2) interaction coefficients, and these, in turn, may modify the shapes of coexistence- and species-area relationships, as well as the coexistence categories of the different species. Perhaps the reviewer is referring to our distinction between "spatial extent" and "environmental heterogeneity". This is a common distinction in the literature on SARs (e.g. references from the main text: Shen et al. 2009, Liu et al. 2020) and, more importantly, it is a conceptually sound way of separating contributions due to increases in area by itself from contributions due to variability in different abiotic conditions.

Third, the estimation

Although statistically sophisticated, statistical sophistication is not without hazards. For instance, it is observational data, and we can expect that local dispersal will make seed distribution patchy. This means that a randomly chosen individual of a species will not in fact have a statistically balanced neighborhood of other species. That leads to a potential bias in the estimation of competition coefficients as has been discussed theoretically by Pacala and Levin 1997. In addition even if estimation is unbiased, the effects on coexistence need not be. For instance, if intraspecific competition is a bit stronger than interspecific competition, as claimed is common in this manuscript, unbiased estimation error will mean in some fraction of cases, the estimate for interspecific competition will be greater than the estimate of intraspecific competition. Thus, even though in reality all species coexist, it will be concluded that some fraction of them do not. They would then be put in the wrong category.

The reviewer raises an important point here. Variability in observed interactions, even if correctly accounted for by the uncertainty captured in the statistical model, may give rise to biased coexistence estimates via small switches in intra-interspecific interaction coefficients. In order to ensure the robustness of our results, we provide in the revised manuscript a new supplementary analysis.

In particular, we performed a sensitivity analysis in which we profit from the uncertainty captured by the Bayesian model. In the main analyses, we use as interaction coefficients the means from the posterior distributions of the statistical model. In the supplementary analyses performed, we repeat the calculation of coexistence-area curves with coefficients drawn randomly from the full posterior distribution of each parameter. We generate 100 replicates for the homogeneous parameterization and 100 replicates for the heterogeneous parameterization, and plot the resulting coexistence-area curves alongside the main ones in Fig. S3 (also included in this response, see below). These analyses show that accounting for the whole variability inherent to the statistical model does not modify our results, as the resulting curves are in all cases similar to the main ones. The deviations we observe (e.g. in year 2017 for the heterogeneous parameterization the replicated curves are systematically richer than the mean one) are slight, involving no more than one or two species of difference at the largest spatial scales. Therefore, this additional analysis reinforces the qualitative interpretation of our main results, and shows that they are robust to incorporating the variability in coefficient estimates obtained with our statistical model.

This analysis is referenced in lines 385–392 of the main text.

We would also like to note that the balanced estimates of species interactions that would be desirable to eliminate all sources of uncertainty are impossible to obtain in experimental studies for communities of more than a modest number of species, as the number of combinations needed grows exponentially with increasing richness. For example, in our system we account for 19 species. In order to obtain perfectly balanced estimates of the effect of each species – by itself and combined –, we would need to experimentally assemble each combination of 1–18 species as neighbors, for all species as focals, which amounts to $19 \cdot 18!$, or approximately $1.2 \cdot 10^{17}$ combinations. The thorough field campaign in practice since 2014 is meant to capture, as much as possible, the interactions actually observed in our study system without reverting to unfeasible experimental designs. Each of the 9 plots has 36 subplots, and the 9 plots are arranged, as explained in the main text, alongside a subtle salinity gradient. Each focal species is measured once in every subplot, and in the case of rare species that do not appear in enough subplots, we sampled more than one individual in the subplots in which they appear. Therefore, we obtain around $36 \cdot 9 = 324$ estimates for each focal species each year. These estimates may indeed be biased as not all species appear together in the field, but our sampling design ensures that these biases are minimized, and if two species frequently interact in the field, we are confident that our scheme captures it. If there is a limited number of observations for a particular species pair, the Bayesian multilevel model will tend to produce estimates that are close to the overall mean, which is a sensible assumption in the absence of better information. Likewise, if an interaction is observed only sporadically, it will have a limited effect on the community-level trends, since for evaluating coexistence patterns we only consider the observed interactions with the frequency with which they appear in the field. Overall, we would like to remark that we are not aware of any other observational study of this type with the number of replicates across time and space that we have obtained. Furthermore, in the unrealistic situation that we were able to assemble all combinations of species needed to obtain better estimates of interaction coefficients, such an experimental design would miss the spatial distribution of field-observed interactions, thus rendering meaningless the obtention of CAR and SAR curves.

Fig. S3. Coexistence-area curves in our study system 1) in color, pointed, using the mean of the posterior distributions of the model parameters (interaction coefficients and intrinsic fecundity rates). These are the same curves and follow the same color scheme as in Fig. 2 of the main text. 2) in light grey, obtained from random draws of the posterior distributions of each parameter. For each year and parameterization, 100 curves are represented.

For these reasons, I see the manuscript as an illustration using a theoretical construct partly based on data from a real system, rather than a strong statement about that system.

We have modified the main text to accommodate the comments of the reviewer, be more explicit about the limitations of the study, and clarify terms such as coexistence of pair of species. We have also changed the title of the manuscript to a less assertive one. Nevertheless, for the reasons exposed above, we believe the interpretation of our results holds, and is robust to different sources of variability (Fig. 4, Fig. S3).

References not in the manuscript

Pacala, S. W., and S. A. Levin. 1997. Biologically generated spatial pattern and the coexistence of competing species, Pages 204-232 in D. Tilman, and P. Kareiva, eds. *Spatial Ecology: The role of space in population dynamics and interspecific interactions*. Princeton, New Jersey, Princeton University Press.

Reviewer #3 (Remarks to the Author):

I am Reviewer 3 of this paper when it was under consideration at Nature. I thank the authors for the detailed and thorough response to my comments. All my key concerns are addressed. I especially appreciate the new binomial GLM testing to validate the (theoretical) classification of transient versus coexistent species. Overall, the manuscript is much easier to follow than the previous version, where all key concepts/methods clearly defined and justified. I applaud the authors for this nice paper and I believe it will be a valuable contribution to the study of SAR and CAR.

We thank the reviewer for his/her kind words and constructive suggestions, which have really contributed to improving the manuscript. We have incorporated the comments below in the updated version.

I only have some very minor points:

- Line 14. ``contrary to expectation" is unclear.

We have removed that wording.

- Line 20 & line 287. The authors highlighted the importance of ``demographic stochasticity" in the Abstract and the importance of ``stochastic and dispersal events" in the Discussion. My understanding is that the authors consider these two things identical (while they are not identical in the general literature). Perhaps the authors should pick one terminology to avoid potential misunderstanding?

The reviewer is right in this appreciation. We have modified the text to avoid confusion and stick with the term "stochastic and dispersal events", in order to account for the spatial component studied.

- Line 52. What does ``showing contrasting population dynamics" mean?

We have changed the term "population dynamics" to "population trajectories".

- Line 55. Refs 14 and 15 studies the feasibility domain solely with competitive interactions (as the structural niche & structural fitness concepts best apply to purely competitive systems).

We have included an additional sentence (Lines 409-410) clarifying that this approach may accommodate different signs of interactions, as discussed in Godoy et al. (2018).

- Line 126. Please remove ``very" from ``very similar".

Corrected

- Line 144. Please remove ``very" from ``very high".

Corrected

- Color usages in the figures are not consistent.

Figures 2 and 4 have the same color scheme (green for richness SAR, dark blue for homogeneous space CAR). We have changed the color scheme of Fig. 3 to avoid confusion with them. For Fig. 1, since it is a conceptual diagram, we chose to use different colors to highlight spatial variability.

- y-axis labels are not consistent across figures. I prefer ``average number of species" in Figure 1C over ``species " in Figs 2-4.

Corrected

Reviewer comments, further round review -

Reviewer #2 (Remarks to the Author):

The authors have gone to considerable efforts to satisfy various concerns, but some important ones remain.

(1) They cite an unpublished paper, Gibbs et al. (2017), that is quite loose in its terminology, and claims to assert that they should not need to check stability of their systems, that feasibility is enough. However, Gibbs et al say that feasibility should be good enough for the matrix M if the interaction matrix A is a stability matrix. So they have to check their interaction matrices for stability at the very least.

(2) Although they acknowledge that they likely have a rich seed bank, they do not acknowledge that differences between species in seed bank dynamics prevent them from accurately determining if the intrinsic growth rates are in the feasibility domain. Thus, despite the heroic empirical work in this study, a key piece of the biology of these species is missing.

(3) They acknowledge that dispersal processes are likely to be important, but say that this would involve another paper to account for them. This is implicit recognition that the present study is not adequate for the conclusions they wish to make. How about acknowledging that?

Reviewer #3 (Remarks to the Author):

I believe that the authors have fully addressed my concerns. The new version has improved substantially in clarity. I don't have any further comment and happy to see it published.

Reviewer #2 (Remarks to the Author):

The authors have gone to considerable efforts to satisfy various concerns, but some important ones remain.

(1) They cite an unpublished paper, Gibbs et al. (2017), that is quite loose in its terminology, and claims to assert that they should not need to check stability of their systems, that feasibility is enough. However, Gibbs et al say that feasibility should be good enough for the matrix M if the interaction matrix A is a stability matrix. So they have to check their interaction matrices for stability at the very least.

We provide in the revised version an analysis of the local stability of the studied communities (Supplementary Note 3), to complement the feasibility analyses of the main text. The reviewer is right in that the Gibbs et al. paper refers to specific configurations of theoretical matrices, and therefore, it is unclear to what extent its insights are applicable to empirical communities. We show that feasibility and local stability are decoupled in our study system, such that virtually no community attains local stability. This has previously been shown to occur in other seasonal systems, in particular plant-pollinator communities (Saavedra et al. 2016, referenced in main text). This analysis is now discussed in the main text (lines 183-187, 417-420) and fully detailed in the Supplementary Material (Supplementary Note 3).

(2) Although they acknowledge that they likely have a rich seed bank, they do not acknowledge that differences between species in seed bank dynamics prevent them from accurately determining if the intrinsic growth rates are in the feasibility domain. Thus, despite the heroic empirical work in this study, a key piece of the biology of these species is missing.

We thank the reviewer for giving us the opportunity to better explain this part of the methodology, that we realize was not fully explained and understandable in previous versions of the manuscript. In the quantification of intrinsic growth rates, seed survival rate and germination rate are actually accounted for. For understanding this, we need to bear in mind the mathematical conditions for the structural framework.

In order to calculate the feasibility of a given system, the system needs to be formulated in terms of a linear Lotka-Volterra model, or an equivalent formulation. The paper by Saavedra et al. (2017, referenced in main text) lays down different model forms that can be adapted to the linear LV formulation, including the one we used in our study, the Ricker annual plant model (see Appendix 4 in Saavedra et al. 2017). There is the particularity, however, that the intrinsic growth rates may not be equivalent between these two formulations, as the annual plant model accounts for seed survival and seed germination rates of the species modelled when calculating the intrinsic fecundity terms (F_i). Thus, intrinsic fecundity rates from the annual plant model (F) need to be converted to intrinsic growth rates in a linear LV formulation (r). For our analyses, we derived the conversion between these two formulations (see new Supplementary Note 2 as well as the R code used to perform this calculation at https://github.com/garciacallejas/SAR_coexistence/blob/main/R/structural_metrics.R).

In that sense, these two aspects of the seed bank (seed survival rate and seed germination rate) are explicitly included in the calculation of the intrinsic growth rates. The seed survival and germination rates are taken from empirical observations from 2016 (see the csv with the obtained values at https://github.com/garciacallejas/SAR_coexistence/blob/main/data/01_05_plant_species_traits.csv). We decided not to detail these intermediate steps in the previous methods section of the manuscript given that 1) they are a purely methodological step that does not add to the conceptual understanding of the methodology, 2) word-limit constraints, and 3) the interested reader can explore in detail these calculations in the included R code. Nevertheless, we realise this may give rise to confusion, as rightly expressed by the reviewer. Therefore, in the revised version of our manuscript we explain in detail this step: 1) in the main methods (lines 379-391), 2) in Supplementary Note 2.

We emphasize that we do not model the dynamics of the seed bank as such, since that is out of the scope of the paper (main text, lines 350-355). However, its contribution is explicitly included in the growth rate calculations. It is also true that we do not have survival rates and germination rates for every year of empirical data. An argument could be made that inter-annual variability in these rates could be important in shaping the feasibility values obtained in our study. In the last submitted version of the manuscript, we performed a thorough sensitivity analysis of the two main sets of parameters obtained from the statistical model: the intrinsic fecundity rates (λ) and the interaction coefficients (α). We now complement this sensitivity analysis with a new one, in which we vary (increase, decrease, and randomize) the observed seed survival and seed germination rates. This new analysis shows that realistic variability in these rates does not substantially vary the resulting coexistence-area relationships. This new sensitivity analysis is included in Supplementary Note 2, Supplementary Figure 4, Supplementary Table 4, and referenced in the main text in lines 103-104, 205-208, 379-391.

(3) They acknowledge that dispersal processes are likely to be important, but say that this would involve another paper to account for them. This is implicit recognition that the present study is not adequate for the conclusions they wish to make. How about acknowledging that?

We now clarify in the main text more explicitly that we do not model dispersal or other spatially explicit mechanisms (main text, lines 334-337), but rather, we use this spatially explicit information to infer coexistence-area relationships. In this view, our results do show that these spatial mechanisms are key for multi-species coexistence in our system, but we still believe that the mechanistic modelling of these spatial mechanisms would be a different study with a different set of research questions, methodologies, and narrative. As we see it, one of the strengths of our study is that it can serve as a mechanistic link between such spatial ecology studies and other types of studies focused on species coexistence and richness patterns.

Reviewer #3 (Remarks to the Author):

I believe that the authors have fully addressed my concerns. The new version has improved substantially in clarity. I don't have any further comment and happy to see it published.

We are really grateful for the comments raised and, importantly, for the overall constructive and positive tone of the reviewer.